# Conformational alteration of DOCK5•ELMO1 signalosome on lipid membrane
Takehiro Shinoda [1], Kazushige Katsura [1], Yoshiko Ishizuka-Katsura[1], Kazuharu Hanada[1], Mayumi Yonemochi[2], Yuki Miyamoto [3], Mutsuko Kukimoto-Niino [1], Junji Yamauchi [4] & Mikako Shirouzu [1,2,5] ✉

The DOCK protein family activates Rho small GTPases through guanine nucleotide exchange factor (GEF) activity. DOCK is thought to exert its GEF activity at the plasma membrane. However, the mechanism by which DOCK activity on the plasma membrane is regulated remains unclear. Herein, we present a new conformation in which DOCK5, ELMO1, RhoG, and Rac1 are aligned on a plane and symmetrically flattened, as revealed by cryo-EM using a lipid membrane-coated grid. The major conformational change leading to this structure results from rotation of each DOCK5•ELMO1 hinge site through interactions with the membrane. Biochemical and cellular experiments indicate that conformational changes driven by acidic lipids are important for regulating the GEF activity of the DOCK5•ELMO1 complex on the plasma membrane and are essential for its downstream signalling. This approach also enables the analysis of large lipid-associated complexes, such as signalosomes, and will aid studies of membrane-dependent signalling assemblies.

Biological membranes play an essential role in small GTPase signalling, serving as sites for their regulation and binding to effectors[1,2]. Rho GTPases, which are molecular switches that regulate cell morphology and migration, are activated and inactivated at the plasma membrane through interactions with guanine nucleotide exchange factors (GEFs) and GTPase-activating proteins (GAPs), respectively. Inactive Rho GTPases are removed from membranes by interacting with Rho GDP dissociation inhibitors (GDIs), which conceal their C-terminal lipidation sites.

The dedicator of cytokinesis (DOCK) protein family acts as a GEF for Rho GTPases[3,4] that participates in various physiological processes, including development and immunity[5,6]. These proteins contain a DOCK-homology region (DHR)-1 domain that mediates membrane association[7] and a DHR-2 domain that facilitates dimerization and the GDP/GTP exchange reaction[8]. Of the 11 proteins in the family, the DOCK-A/B sub-family proteins (DOCK1/2/3/4/5) bind to engulfment and cell motility (ELMO) scaffold proteins; the DOCK•ELMO complex functions as a GEF for Rac[9]. The current model suggests that DOCK1[10], ELMO1[11], and their complexes are autoinhibited in their basal state[5]. The autoinhibition of the DOCK1•ELMO complex is lifted when ELMO binds to active GTPases such

as RhoG[12], Arl4A[13], and other membrane-associated proteins that are activated via cell stimulation.

The cryo-electron microscopy (cryo-EM) structures of the DOCK2•ELMO1 and DOCK5•ELMO1 complexes have revealed a common mechanism of autoinhibitory regulation[14,15]. These complexes undergo large conformational changes (from closed to open) to achieve the Rac1-bound conformation[14,16]. This is facilitated by active RhoG, whose structural basis is shown by the RhoG-bound DOCK5•ELMO1•Rac1 complex structure[15]. Although cryo-EM analyses of these simplified soluble systems have revealed details of the inactive and active conformations of the DOCK2/5•ELMO1 complex, the conformation of the DOCK•ELMO complex exhibiting GEF activity on the membrane remains unknown. In the reported DOCK2/5•ELMO1•Rac1 complex structures, multiple membrane-binding elements (i.e., the DHR-1 domain of DOCK and the C-terminus of Rac1) are not aligned in the two-fold symmetric dimer. This suggests that the DOCK•ELMO complex may undergo some conformational changes during membrane binding or bind asymmetrically to the membrane to activate Rac1[2]. Hence, we aimed to determine the structure of the DOCK5•ELMO1 complex on phospholipid membranes.

[1]Laboratory for Functional and Structural Biology, RIKEN Center for Integrative Medical Sciences, Yokohama, Kanagawa, Japan. [2]Drug Discovery Structural Biology Platform Unit, RIKEN Center for Integrative Medical Sciences, Yokohama, Kanagawa, Japan. [3]Laboratory of Molecular Pharmacology, National Research Institute for Child Health and Development, Tokyo, Japan. [4]Laboratory of Molecular Neurology, Tokyo University of Pharmacy and Life Sciences, Tokyo, Japan. [5]Structural Life Sciences and Cell Biology Collaboration Team, RIKEN Center for Biosystems Dynamics Research, Yokohama, Kanagawa, Japan. ✉e-mail: mikako.shirouzu@riken.jp

## Results

### Cryo-EM for DOCK signalosome on membrane

To begin the structural analysis of the DOCK signalosome on the lipid membrane, we first attempted to reconstitute the DOCK5•ELMO1 complex with lipid-modified Rac1 on the lipid membranes of small unilamellar vesicles (SUVs). Using cryo-EM, we successfully observed that the DOCK5 complex attached to the lipid membranes of SUVs. However, the non-uniform curvature of the lipid membrane affected the DOCK5 complex structure on the lipid membrane, preventing us from obtaining convergent three-dimensional particle images in our single-particle analysis (Supplementary Fig. 1a−d). To overcome this problem, we developed a method based on preciously used techniques[17,18] to reconstitute DOCK signalosomes by attaching lipid monolayers to a cryo-EM grid (Fig. 1). The human DOCK5 and ELMO1 complex, along with an upstream signalling factor, a constitutively active mutant of RhoG$^{Q61L}$ tagged with polyhistidine at the C-terminus, and a substrate, nucleotide-free mutant Rac1$^{G15A}$ also tagged with a poly His-tag at its C-terminus (Fig. 2a), were reacted on a grid coated with a lipid membrane [67PC/21PA/6PIP3/6DOGS-NTA Ni (mol%)], followed by cryo-EM data collection. For the single-particle analysis, we obtained a 3D map with an angular contribution of approximately 45° by merging the cryo-EM data measured from the top of the sample grid and from a tilt angle of 30° (Supplementary Fig. 2).

Figure 2b shows the results of 2D classification from the cryo-EM single-particle analysis of this sample. Reflecting binding to the lipid membrane, we observed approximately all particles of the DOCK5•ELMO1 complex, appearing S-shaped with an extended longitudinal axis, in a uniform orientation. Subsequent 3D reconstruction of the cryo-EM map and structural modelling revealed that these membrane-bound particles adopted a shape markedly different from the semi-arc conformation of the auto-inhibited DOCK5•ELMO1 complex (PDB ID: 8JHK), also referred to as the auto-inhibited closed form, as reported by Kukimoto-Niino et al.[15]. (Fig. 2c, d). The ~7 Å resolution map, particularly in the regions corresponding to the armadillo repeat motif (ARM) and DHR-2 domains of DOCK5, exhibited sufficient quality to trace the Cα chains (Fig. 2c, Supplementary Fig. 3, and Table 1). Other poorly mapped sites, notably the N-terminal domains (NTD) of ELMO1 and RhoG, were visualized relatively clearly through local refinement (Supplementary Fig. 4). We then modelled the entire complex molecule using the reported structures[15,16,19]. ELMO1NTD, which interacted with DHR-2 via the Ras-binding domain (RBD) in its auto-inhibited closed form (Fig. 2d, right), diverged from DHR-

2 and expanded along the lipid membrane surface due to the binding of RhoG to the RBD (Fig. 2d, left). This large conformational change brought DHR-2 into close proximity to the lipid membrane and allowed it to bind to the membrane-anchored Rac1. The Pleckstrin homology (PH) domain in the C-terminal domain (CTD) of ELMO1, which faces lobe B of DHR-2 in reported cryo-EM structures[15,16], was oriented in the same direction as DHR-1, including the PIP3 binding site[20,21], near to the lipid membrane (Fig. 2d, left). We refer to this conformation, in which each molecule is in close proximity to the lipid membrane, as the extended-open form.

### Conformational change is required for GEF activity and downstream signalling

Next, we studied the large conformational changes from the closed to the extended-open form in detail. Figure 3a shows the closed and extended-open forms superimposed on the DHR-2 domain of DOCK5. The conformational change from the closed to the extended-open form showed that DOCK5 rotated ~20° around the DOCK5 hinge site (Glu1215) connecting the ARM and DHR2 domains (Supplementary Fig. 5a) and elongated in the long-axis direction; whereas, we observed no clear conformational changes in other portions of DOCK5. ELMO1CTD, which is tightly bound to the SH3 and helical region of DOCK5, also rotated along with DOCK5 such that the variable loop connecting strands β1 and β2 (VL1) on the ELMO1 PH domain (Supplementary Fig. 5b) was oriented toward the lipid membrane. Simultaneously, the β3 and β4 strands of the PH domain where Lys584 and Lys607 were positioned (Supplementary Fig. 5b) were also in close proximity to the DOCK5 hinge site, where Asp1214 and Glu1215 were positioned (Fig. 3b, c). The basic amino acids Arg563, Arg568, Arg569, Arg570, and Lys573 in the VL1 region were in close proximity to the lipid membrane. Notably, two basic residues (Lys584 and Lys607) adjacent to the DOCK5 hinge are conserved among ELMO subtypes (Supplementary Fig. 5b). Moreover, two acidic residues within the DOCK5 hinge region are conserved in the DOCK-A/B subfamily (Supplementary Fig. 5a). In addition, five basic residues within the VL1 loop of the PH domain, positioned adjacent to the lipid membrane, are also conserved among ELMO subtypes (Supplementary Fig. 5b).

We first investigated the involvement of these residues in membrane binding. Figure 4a shows the results of the interaction assays of purified ELMO1CTD with lipid molecules. Wild-type ELMO1CTD interacted with acidic phospholipids (mainly PA and PS), whereas Ala mutants of these five basic amino acids (R563A/R568A/R569A/R570A/K573A) (hereafter

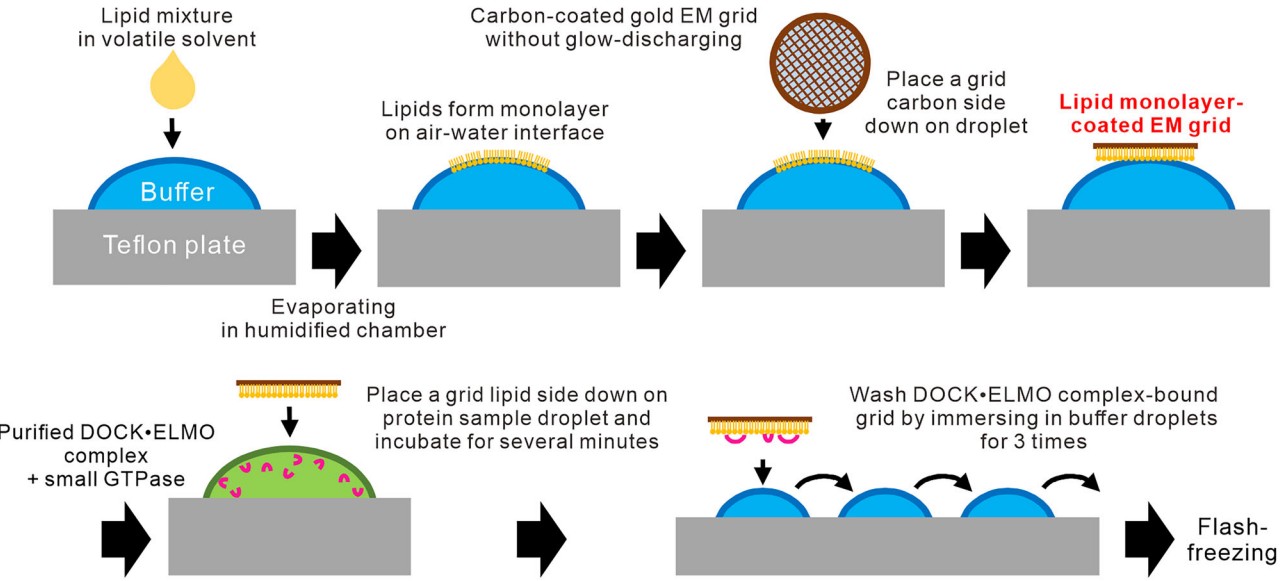

**Fig. 1 | Preparation of cryo-EM sample of DOCK•ELMO complex on lipid monolayer-coated EM grid.** DOCK•ELMO complexes and small GTPases were reconstituted on lipid monolayers adsorbed on the carbon-coated side of a cryo-EM grid. See Materials and Methods for details.

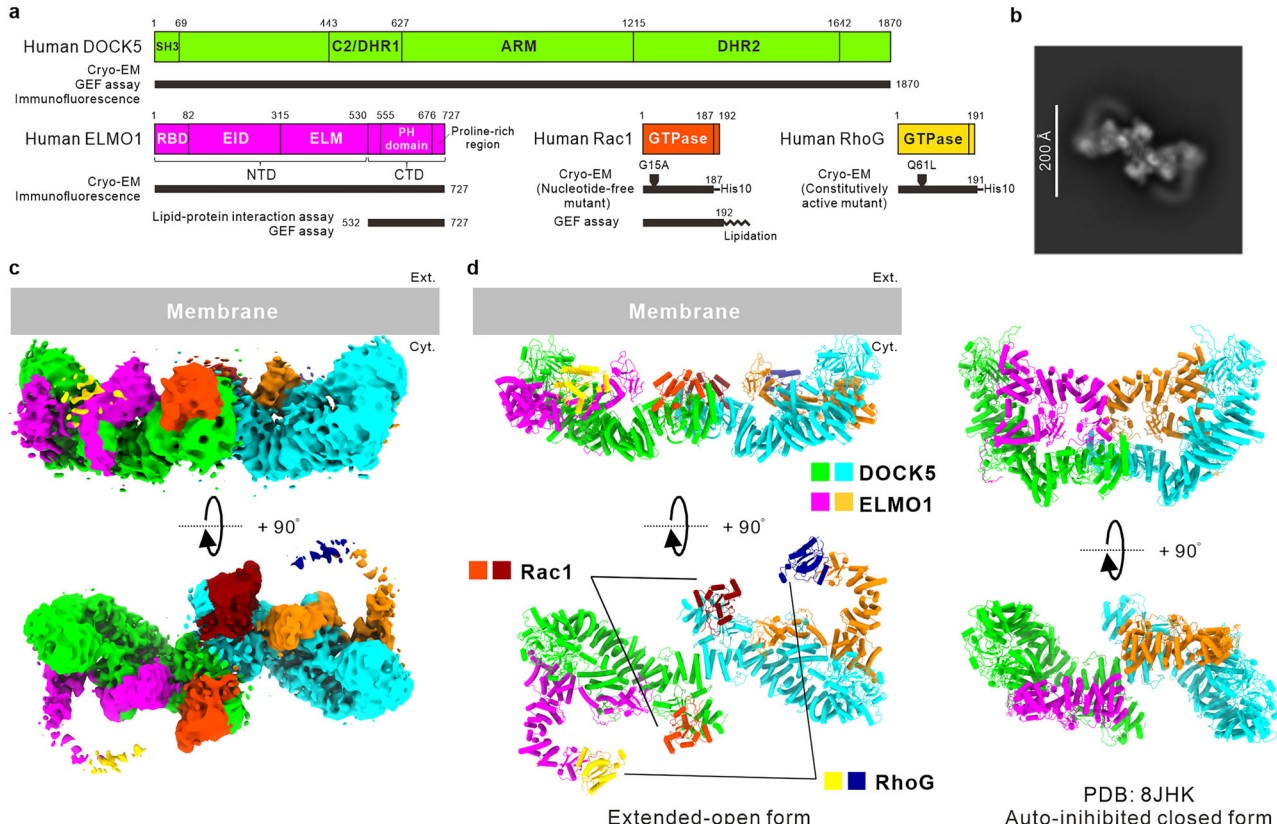

**Fig. 2 | Cryo-EM structure of DOCK5•ELMO1 complex with Rac1 and RhoG on a lipid membrane. a** Domain organisation of human DOCK5, ELMO1, and two Rho-GTPases, Rac1 and RhoG, and of the protein constructs used in each experiment in this study (black bar). DHR Dock homology region, ARM armadillo repeat domain, RBD Ras-binding domain, EID ELMO inhibitory domain, ELM ELMO domain. **b–d** Cryo-EM single-particle analysis of the DOCK5•ELMO1 complex bound to Rac1 and RhoG. Shown are the 2D class average (**b**), cryo-EM density maps displayed at a contour level of 0.12 (**c**), and the cryo-EM structural model of the extended-open form of the DOCK5•ELMO1 complex associated with Rac1 and RhoG, generated from the maps in (**c**) and compared with the auto-inhibited closed form of the DOCK5•ELMO1 complex (PDB: 8JHK) (**d**). In panels (**c**) and (**d**), the upper and lower images represent front and top views, respectively.

referred to as the VL1 mutant) did no bind to acidic lipids (Fig. 4a). We further examined this interaction using a coprecipitation assay with SUVs. In the presence of PA, the amount of ELMO1CTD coprecipitating with SUVs increased, whereas the VL1 mutant exhibited a reduced level of coprecipitation compared with that of the wild-type (Fig. 4b, c). These results suggested the presence of electrostatic interactions between the basic amino acid residues in VL1 and the acidic phospholipids in lipid membranes. Incidentally, Chang et al.[14] reported that in the cryo-EM structure of the DOCK2•ELMO1•Rac1 complex in solution, ELMO1NTD was largely rotated and opened similar to the extended-open form observed in the current study. However, the RBD was oriented in the opposite direction relative to the PIP3 binding site of DHR-1. In addition, the interaction of the PH domain of ELMO1 with Rac1 and lobe B of DHR-2 was preserved, whereas the conformational change due to the rotation of the DOCK2 hinge was not sufficient to form a flattened conformation, differing from the extended-open form observed in the present study (Supplementary Fig. 6).

Next, to evaluate the interactions observed in the extended-open form, we examined the GEF activity of various DOCK5•ELMO1CTD complexes (wild-type and mutant) on the membrane (Fig. 4d, e). We performed GEF assays in the presence of liposomes containing PC, PA, and PIP3, with lipid-modified Rac1 complexed with RhoGDI as the substrate. Lipid-modified GDP-bound Rac1 binds tightly to RhoGDI. In this state, exchange with GTP does not proceed; however, interaction with the GEF protein on lipid membranes induces the dissociation of RhoGDI, thereby allowing exchange with GTP[1,22,23]. Likewise, in our study, we observed no GTP exchange activity in the absence of either SUVs or the GEF protein, DOCK5•ELMO1 complex, whereas activity was clearly detected only when both were present (Fig. 4d). This indicated that the assay specifically detects the GEF activity of

the DOCK5•ELMO1 complex on the membrane. Using this assay system, we assessed the GEF activity of various mutants, as shown in Fig. 4e. The GEF activity of the complex consisting of wild-type DOCK5, DOCK5^WT, and the VL1 mutant of ELMO1CTD, which lacks the ability to bind to acidic phospholipids (Fig. 4a−c), was approximately 90% of that of the wild-type DOCK5•ELMO1CTD complex, indicating a slight reduction in activity. Conversely, the complex of DOCK5^WT with the Ala mutant of K584/K607 on the β3 and β4 strands of the PH domain (hereafter known as the β3/β4 mutant) located in close proximity to the DOCK5 hinge (Fig. 3b), showed no decrease in GEF activity. The complex of the Ala mutant of D1214/E1215 at the DOCK5 hinge site (hereafter referred to as the hinge mutant) with the wild-type ELMO1CTD, ELMO1CTD^WT, also showed no decrease in GEF activity. In contrast, the complex of the hinge mutant of DOCK5 with the β3/β4 mutant of ELMO1CTD showed a significant decrease in GEF activity, ~70% compared with that of the DOCK5^WT•ELMO1CTD^WT complex (Fig. 4e). These results suggested that, among the interactions formed through the conformational transition into the extended-open form, the interaction between the DOCK5 hinge and the β3 and β4 strands of the PH domain is involved in GEF activation, whereas the interaction between the VL1 region of the PH domain and the lipid membrane is less significantly involved.

We examined the effect of ELMO1 mutations on lamellipodia formation by observing localised filamentous actin formation, which underlies lamellipodia structures, in COS-7 cells via immunofluorescence staining (Fig. 5a−c). The expression levels of wild-type and mutants of ELMO1 transfected into COS-7 cells were comparable (Fig. 5b). Even in cells not transfected with DOCK5 or ELMO1, we observed peripheral localisation of filamentous actin and lamellipodia formation, with serum (FBS) stimulation

**Table 1 | Cryo-EM data collection, refinement, and validation statistics**

| | DOCK5•ELMO1 complex with RhoG and Rac1 Grobal(EMDB-63464) (PDB 9LX0) | DOCK5•ELMO1 complex with RhoG and Rac1 Local (EMDB-63477) (PDB 9LXH) |
|---|---|---|
| Data collection and processing | | |
| Magnification | 64,000 | |
| Voltage (kV) | 300 | |
| Electron exposure (e⁻/Å²) | 50 | |
| Defocus range (μm) | 0.8–2.0 | |
| Pixel size (Å) | 1.33 | |
| Symmetry imposed | $C2$ | $C1$ |
| Initial particle images (no.) | 510,098 | 510,098 |
| Final particle images (no.) | 55,365 | 55,365 |
| Map resolution (Å) | 6.98 | 7.52 |
| FSC threshold | 0.143 | 0.143 |
| Map resolution range (Å) | 6.12–16.8 | 6.04–15.6 |
| Refinement | | |
| Initial model used (PDB code) | 8JHK, 7DPA, 7Y4A | 9LX0 |
| Model resolution (Å) | 8.91 | 8.43 |
| FSC threshold | 0.5 | 0.5 |
| Map sharpening $B$ factor (Å²) | −583.6 | −812.0 |
| Model composition | | |
| Non-hydrogen atoms | 44,804 | 17,437 |
| Protein residues | 5452 | 2162 |
| Ligands | 0 | 0 |
| $B$ factors (Å²) | | |
| Protein | 275.00 | 95.00 |
| R.m.s. deviations | | |
| Bond lengths (Å) | 0.002 | 0.002 |
| Bond angles (°) | 0.517 | 0.575 |
| Validation | | |
| MolProbity score | 2.18 | 2.66 |
| Clashscore | 11.86 | 16.62 |
| Poor rotamers (%) | 3.16 | 5.66 |
| Ramachandran plot | | |
| Favoured (%) | 96.67 | 94.71 |
| Allowed (%) | 3.33 | 5.29 |
| Disallowed (%) | 0.00 | 0.00 |

further enhancing this localisation (Vector in Fig. 5a, c). In cells transfected with DOCK5^WT and ELMO1^WT, this phenotype was more pronounced, and after serum stimulation, we observed localised lamellipodia in approximately all cells examined. Furthermore, the expressed ELMO1 was localised at the cell periphery together with actin, and appeared to be enriched in response to serum stimulation (WT in Fig. 5a, c). Conversely, in cells transfected with the β3/β4 mutant of ELMO1, peripheral localisation of actin and lamellipodia formation were reduced, with serum stimulation

resulting in limited response (β3/β4 mutant in Fig. 5a, c). The VL1 mutant of ELMO1 resulted in an even more pronounced suppressive phenotype, with minimal actin accumulation or lamellipodia formation at the cell periphery. This pronounced suppression observed with the VL1 mutant suggested a possible dominant-negative effect (VL1 mutant in Fig. 5a, c).

## Schematic diagram of the conformational changes in DOCK5•-ELMO1 signalosome on lipid membrane

A schematic diagram of the conformational changes in the DOCK5•-ELMO1 complex on the lipid membrane predicted from our results is shown in Fig. 6. In this schematic diagram, the DOCK5•ELMO1 complex in the cytoplasm attaches to the lipid membrane via the interaction of the DOCK5_DHR-1 domain with PIP3, which accumulates on the inner leaflet side in response to external stimuli. Given that the upstream signalling factor RhoG is also localised to the inner leaflet of membrane domains enriched in acidic phospholipids, including PIP3, and has been reported to contribute to the membrane recruitment of DOCK•ELMO complex[12], in this study, RhoG and DOCK5•ELMO1 complex may have been colocalised within shared membrane domains (Step 1 in Fig. 6). Next, the DOCK5•ELMO1 complex, which is in equilibrium between the autoinhibited closed and open form, in which the RBD of ELMO1 dissociates from DOCK5_DHR-2[15], shifts its equilibrium to the open form through the interaction of the RBD with RhoG on the lipid membrane. Although not addressed in this study, another upstream factor, the G protein-coupled receptor BAI1, may shift the equilibrium toward the open form through interaction between its cytoplasmic C-terminal region and the EID region of ELMO1[14,24] (Step 2 in Fig. 6). The basic residues in the VL1 of the ELMO1_PH domain are exposed in the open form, allowing them to interact with acidic lipids on the lipid membrane (Step 3 in Fig. 6). As the ELMO1_PH domain approaches the lipid membrane for this interaction, a major conformational change occurs around the DOCK5 hinge, resulting in an extended-open form. Based on prior analyses, the PH domain interacts with acidic phospholipids in the membrane through the VL1 loop, which is enriched in basic residues (Fig. 4a−c). Therefore, in the extended-open form, acidic lipids are likely concentrated around the PH domain located near the lipid membrane. Furthermore, as shown in Fig. 4d, in the presence of both the lipid membrane and DOCK5•ELMO1 complex, Rac1 dissociates from RhoGDI and undergoes nucleotide exchange to adopt the GTP-bound form. Hence, the DOCK5•ELMO1 complex may undergo a conformational change to the extended-open form, bringing the DHR-2 domain closer to the lipid membrane, thereby facilitating the dissociation of the RhoGDI•Rac1 complex and enabling Rac1 to interact with DHR-2 at the GEF catalytic site of DOCK5 (Step 4 in Fig. 6).

## Discussion

In this study, we developed a method to visualize the conformation of DOCK5•ELMO1 complexes bound to lipid membranes by employing lipid membrane-coated cryo-EM grids (Fig. 1). Utilizing this approach, we determined the unique conformation of the DOCK5•ELMO1 complex with Rac1 and RhoG on lipid membranes and uncovered part of the activation mechanism involving large conformational changes that occur on lipid membranes (Fig. 2c, d). This feature is likely conserved among DOCK-A/B family members that require ELMO for downstream signalling, as the amino acid residues responsible for this conformational change are conserved among these members (Supplementary Fig. 5a, b). Furthermore, this mechanism of conformational change via interaction with lipid membranes also explains the formation of the DOCK5•ELMO1 complex diprotomer. When the DOCK5•ELMO1 complex interacts with the lipid membrane as a monoprotomer, it fails to form a semi-arc structure on the membrane, and consequently, the interaction between the lipid membrane and either ELMO1NTD or the PH domain likely is not coupled to the conformational change of the DOCK5•ELMO1 complex.

First, we discuss the structural role of RhoG in activating the DOCK5•ELMO1 complex. The upstream factor, RhoG, is reportedly one of the factors[25,26] that promotes GEF activity by inducing conformational

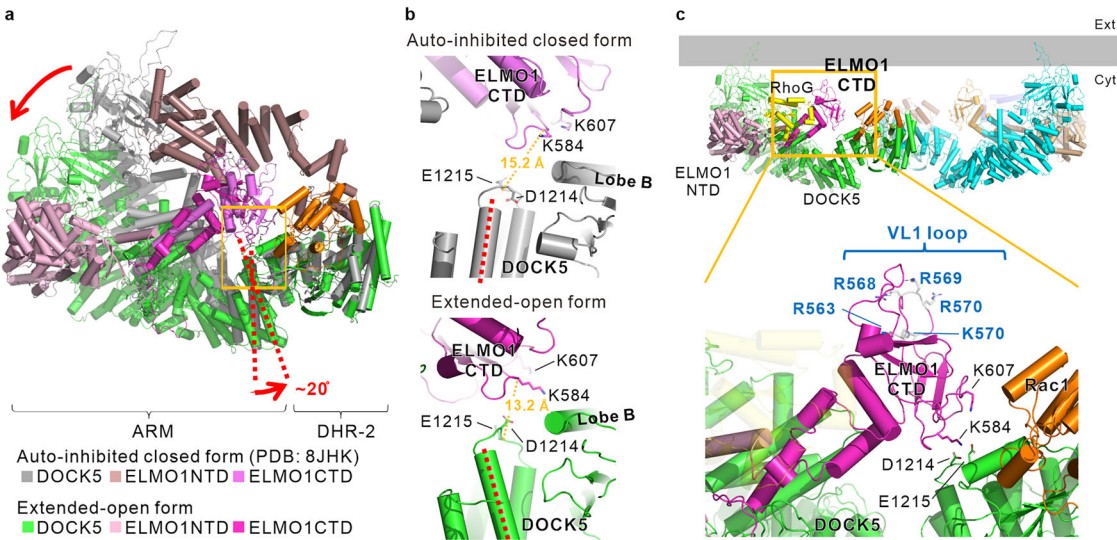

**Fig. 3 | Interactions due to the conformational change of the complex formed by DOCK5•ELMO1 and two Rho-GTPases in the extended-open form. a** Structural models of the closed (PDB: 8JHK) and extended-open forms of the complex superimposed on the DHR-2 domain. Movements of the α-helix (residues 1186–1211) at the C-terminus of the ARM domain, which connects to the DHR-2 domain in DOCK5 and serves as the epicentre of the conformational change, are indicated by red dashed lines. **b** Conformational change in the region highlighted by the orange box in (**a**) and its effect on the interaction between DOCK5 and ELMO. The transition to the extended-open form brings the acidic residues D1214 and E1215 in the hinge region of DOCK5 (between the ARM and DHR-2 domains) into close proximity with the basic residues K584 and K607 in the PH domain of ELMO1. **c** PH domain of ELMO1 oriented toward the lipid membrane in the extended-open form. Basic residues on the VL1 loop of the PH domain are positioned close to the lipid membrane.

changes in the DOCK5•ELMO1 complex. However, the dissociation constant of RhoG for the RBD is $\sim 10^{-5}$ M[14,15], indicating relatively weak affinity. The lipid-modified active form of RhoG should be located on the plasma membrane to activate the DOCK5•ELMO1 complex. The closed form of the DOCK5•ELMO1 complex reported by Kukimoto-Niino et al.[15] suggested that the RBD is located away from the membrane; therefore, RhoG on the membrane is unlikely to interact with the RBD in the closed form to play an active role in separating it from DHR-2. In addition, ELMO1NTD not only contains numerous phosphorylation sites but also interacts with the C-terminal region of BAI1, both of which are involved in regulating GEF activity[14,24]. Therefore, as observed in the present study, RhoG may stabilize the extended-open form by promoting a series of conformational changes in the positive direction by holding the dissociated NTD from DHR-2 close to the membrane through interaction with the RBD. In addition, various factors other than RhoG may be involved in regulating the equilibrium between the closed and open forms, finely regulating GEF activity.

Next, we discuss the functional implications of the interaction between the ELMO1_PH domain and the lipid membrane in the extended-open form. The direct interaction between the PH domain of ELMO1 and the lipid membrane, observed in the extended-open form of the DOCK5•ELMO1 complex and first elucidated in this study (Fig. 3c), represents a notable feature of this conformational transition. Another interaction observed in the extended-open form, between the DOCK5 hinge region and the β3/β4 segment of ELMO1CTD (Fig. 3b), led to reduced GEF activity upon mutation (Fig. 4e) and was also associated with a suppressive effect on lamellipodia formation (Fig. 5a, c). These results suggested that the newly identified interaction involving the DOCK5 hinge region contributes to the stabilization of this conformation. In contrast, VL1 loop mutations in the PH domain, which engages with the lipid membrane, exerted a strong dominant-negative-like effect on actin accumulation at the cell periphery and lamellipodia formation (Fig. 5a, c), while having minimal effect on GEF activity (Fig. 4e). This suggested that the interaction between the PH domain and the lipid membrane plays only modest role in the transition into the extended-open form and instead primarily functions in downstream signal transduction. Focusing on the involvement of the lipid membrane, acidic phospholipids have been reported to promote the dissociation of RhoGT-Pases from RhoGDI[27,28]. In addition, GTP-bound and active mutants of

Rac1, such as Rac1[G12V], reportedly cluster on lipid membranes in the presence of the acidic lipids PA and PIP3. The basic residues in the C-terminal region of Rac1, which interact with these acidic lipids, play important roles in downstream signal transduction[29,30]. In the extended-open form revealed in the current study, Rac1 on DHR-2 and the PH domain are in close proximity to the lipid membrane (Fig. 2d and Step 4 in Fig. 6). As shown in Fig. 4a−c, the interacting partner of the basic residues on the VL1 of the PH domain is an acidic lipid, suggesting that both Rac1 and ELMO1_PH domains likely share a local lipid membrane environment rich in acidic lipids, such as PA and PIP3. Therefore, based on these findings and previous studies, we propose that the PH domain in the extended-open form of the DOCK5•ELMO1 complex likely functions to organise a localised membrane environment enriched in acidic phospholipids. In a manner similar to that depicted in Step 4 of Fig. 6, this localised lipid remodelling is expected to extend to the area surrounding the catalytic domain DHR-2 on the membrane, thereby promoting the dissociation of Rac1 from RhoGDI. The released Rac1 then binds to the catalytic DHR-2 domain, where it undergoes nucleotide exchange and becomes activated. Once activated, Rac1 dissociates from DHR-2 and clusters within the same acidic lipid-enriched membrane domain, thereby contributing to downstream signal transduction. These features may represent core elements of the DOCK5•ELMO1 signalosome. To examine this hypothesis, determining when during the signalling process Rac1 undergoes clustering on the membrane and characterizing the accompanying local lipid distribution, ideally through super-resolution microscopy or cryo-electron tomography for high-resolution structural analysis in situ, will be essential.

In conclusion, the extended-open form, one of the conformations of the DOCK5•ELMO1 complex on lipid membranes determined in the current study, revealed that a major conformational change accompanied by interactions with acidic lipids is required for efficient GEF activity of the DOCK5•ELMO1 complex on lipid membranes. Furthermore, the upstream factor RhoG on the plasma membrane plays a role in promoting and maintaining the conformational change to the extended-open form by binding to the NTD of ELMO1. These results represent just one aspect of the intricate relationship between lipid membranes and the signalling mechanism of the DOCK5•ELMO1 complex. Under the current experimental conditions, whether membrane binding itself or the subsequent

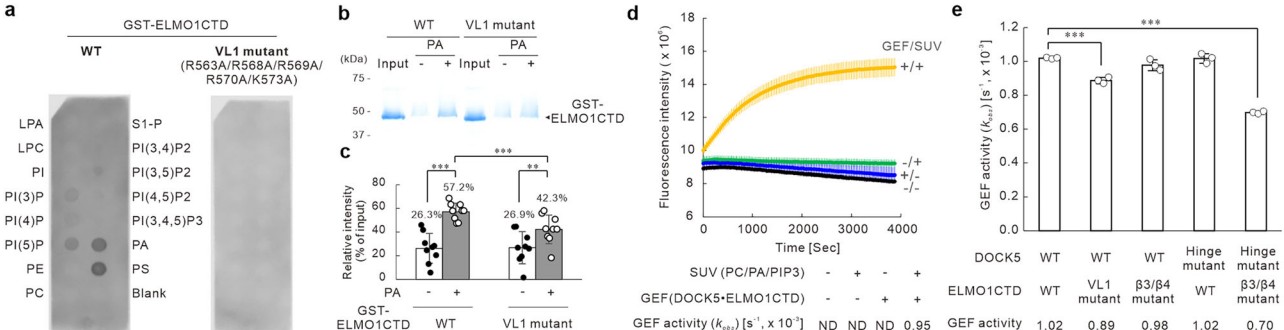

**Fig. 4 | Mutation in ELMO1 reduces its lipid-binding ability and decreases GEF activity of DOCK5. a** Lipid–protein interaction assay for ELMO1CTD. Proteins were detected using an anti-GST antibody. **b** Coprecipitation assay using small unilamellar vesicles (SUVs). ELMO1CTD that coprecipitated with SUVs via centrifugation at $100,000 \times g$ were detected using SDS–PAGE followed by Coomassie Brilliant Blue (CBB) staining. **c** Densitometric analysis of the coprecipitated ELMO1CTD shown in (**b**). Data are presented as the mean ± SD ($n = 9$; unpaired two-sided Student's $t$ test, ***$p < 0.005$, **$p < 0.01$, *$p < 0.05$). **d** GEF assay using

RhoGDI•Rac1 to evaluate the GEF activity of DOCK5•ELMO1CTD in the absence and presence of SUVs. Graph shows the average of three experiments, with error bars representing standard deviation (SD). **e** Assessment of GEF activity of DOCK5•ELMO1CTD mutants. GEF activity was assessed by measuring the exchange of GDP for Bodipy-GTP on Rac1 in the presence of SUVs composed of 78PC/20PA/2PIP3 (mol%). Data are presented as the mean ± SD ($n = 3$; unpaired two-sided Student's $t$ test, ***$p < 0.005$).

structural rearrangement (e.g., adoption of the extended-open form) is the rate-limiting step, particularly in the presence of ELMO1, remains unclear. We acknowledge this limitation and propose that future experiments will be necessary to determine the sequence and identity of the rate-limiting step. In addition, further investigation into the behaviour of small GTPases and related molecules in the DOCK signalosome on plasma membranes is warranted.

## Materials and methods
### Materials
All chemicals for protein sample preparation were purchased from Nacalai Tesque (Kyoto, Japan). ʟ-α-phosphatidylcholine from egg yolk (840041), 1,2-dioleoyl-sn-glycero-3-phosphoethanolamine (850725), ʟ-α-phosphatidic acid from egg yolk (840101), and 18:1 DOGS-NTA(Ni) (790404) were purchased from Avanti Polar Lipids (Birmingham, AL, USA). Phosphatidylinositol 3,4,5-triphosphate diC16 (P-3916) was purchased from Echelon Biosciences (Salt Lake City, UT, USA). cDNA clones encoding human DOCK5, ELMO1, Rac1, and RhoG have been described previously[15,16].

### Protein preparation
Genes encoding human RhoG (residues 1–191, Q61L mutant) and Rac1 (residues 1–187, G15A mutant) were cloned into a pCR2.1 expression vector (Invitrogen, Carlsbad, CA, USA) with a modified natural polyhistidine (N11, MKDHLIHNHHKHEHAHAEH) tag followed by a tobacco etch virus (TEV) protease cleavage site at the N-terminus and a decahistidine (His10) tag at the C-terminus. Each protein was expressed using an *Escherichia coli* cell-free protein synthesis system[15,31,32] and purified using HisTrap column chromatography (Cytiva, Marlborough, MA, USA). The N-terminal N11 tag was cleaved using TEV protease. The resulting C-terminal His10-tagged proteins were further purified using a second HisTrap column chromatography, followed by size-exclusion chromatography on a HiLoad 16/600 Superdex 75 pg column (Cytiva). To purify the active form of RhoG$^{Q61L}$, 10 μM GTP and 1 mM MgCl$_2$ were added during the process. The final buffer contained 20 mM HEPES-NaOH (pH 7.5), 300 mM NaCl, 1 mM MgCl$_2$, and 2 mM DTT. The RhoGDI•Rac1 complex used in the GEF assay was prepared using a mammalian cell expression system. The gene encoding human RhoGDI (residues 1–204) was cloned into the mammalian expression vector pOriP[33] harbouring N-terminal FLAG and streptavidin-binding peptide (SBP) tags, followed by a TEV protease cleavage site. Similarly, human Rac1 (residues 1–192) was cloned into a pOriP vector but with a His10 tag instead of FLAG and SBP tags. RhoGDI and Rac1 were co-expressed in 293 C18 cells (ATCC, Manassas, VA, USA) using the 293 fectin

transfection reagent (Life Technologies, Carlsbad, CA, USA) according to the manufacturer's protocol. Cells were grown in FreeStyle 293 Expression Medium (Thermo Fisher Scientific, Waltham, MA, USA) using an incubation shaker equipped with a CO$_2$ controller (TAITEC Corporation, Saitama, Japan) and harvested 48 h after transfection. Thereafter, the cells were re-suspended in lysis buffer [20 mM Tris-HCl (pH 8.0), 300 mM NaCl, 2.5 mM MgCl$_2$, 5 μL DNase] and then disrupted via sonication. Subsequently, the lysate was cleared via centrifugation at $100,000 \times g$ at 4 °C. The RhoGDI•Rac1 complex was purified from the supernatant using affinity tags on each protein, first using a HisTrap column and then Streptavidin Sepharose HP resin (Cytiva). Next, the affinity tags for each protein were truncated using TEV protease and removed using size-exclusion chromatography on a HiLoad 16/60 Superdex 200 pg column (Cytiva) equilibrated with 20 mM HEPES-NaOH (pH 8.0), 150 mM NaCl, 5 mM MgCl$_2$, and 2 mM DTT. The Tag-truncated purified RhoGDI•Rac1 complex was eluted in a mono-disperse peak and then collected.

The human DOCK5•ELMO1 complex was prepared as previously described[16]. Briefly, full-length DOCK5 (residues 1–1870) with an N-terminal FLAG and SBP tags and full-length ELMO1 (residues 1–727) with an N-terminal FLAG tag were co-expressed in FreeStyle™ 293-F cells (Thermo Fisher Scientific). The resulting DOCK5•ELMO1 complex was purified using Streptavidin Sepharose beads (Cytiva). After cleavage of the N-terminal tag using the TEV protease, the complex was further purified using size-exclusion chromatography on a HiLoad 16/600 Superose 6 pg column (Cytiva) in 20 mM HEPES-NaOH (pH 8.0), 300 mM NaCl, and 1 mM TCEP. The DOCK5•ELMO1CTD complex for the GEF assay was prepared in the same manner as the DOCK5•ELMO1 complex, without the N-terminal tag truncation. For the lipid–protein interaction assay, the gene encoding human ELMO1CTD (residues 532–727) was cloned into a pCR2.1 expression vector with an N11 tag and glutathione S-transferase (GST) at the N-terminus. ELMO1CTD was expressed in an *E. coli* cell-free protein synthesis system. The synthesized GST-tagged ELMO1CTD was adsorbed onto Ni-NTA superflow resin (QIAGEN, Hilden, Germany) prewashed with wash buffer [20 mM imidazole, 20 mM Tris-HCl (pH 8.0), 800 mM NaCl] and then eluted with elution buffer [500 mM imidazole, 20 mM Tris-HCl (pH 8.0), 500 mM NaCl]. Afterward, the eluted fractions were pooled and finally dialysed against 20 mM Tris-HCl (pH 8.0), 150 mM NaCl, and 2 mM DTT overnight at 4 °C.

### Membrane-coated cryo-EM grid preparation and data acquisition
Membrane-coated grids were prepared using a simplified method based on the technique by Levy and Shen[17,18]. All procedures were performed at

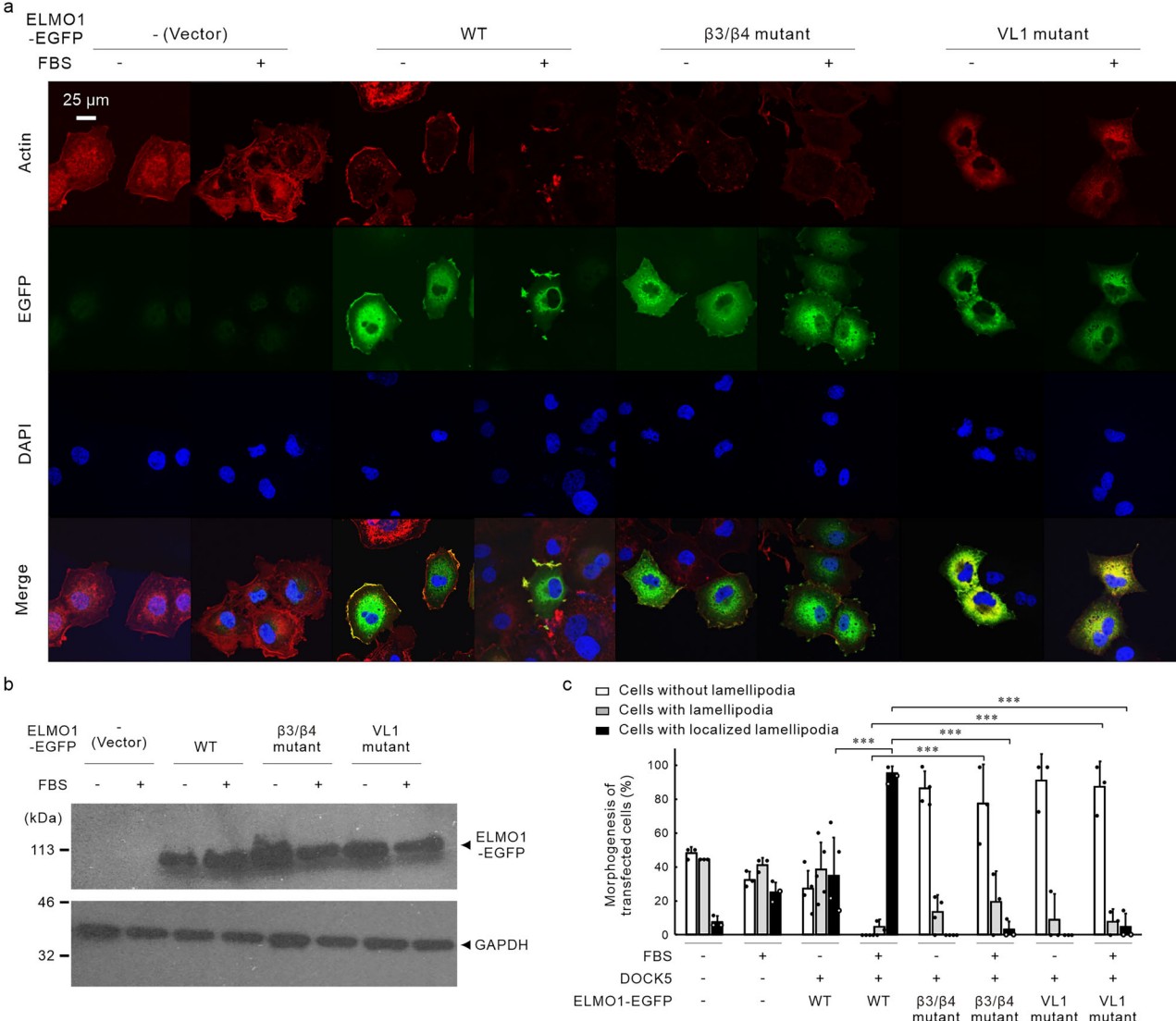

**Fig. 5 | Mutated ELMO1 decreases localised filamentous actin formation. a** COS-7 cells were transfected with a plasmid encoding DOCK5 with or without wild-type or mutant ELMO1 and stimulated with 10% FBS (FBS+) or serum-free medium (FBS−). Transfected cells were stained using an anti-GFP antibody (to detect ELMO1) pre-loaded with green fluorescence dye-conjugated secondary antibody and red fluorescence dye-conjugated phalloidin (to detect filamentous actin). Cells were finally mounted using a DAPI-containing mounting medium (blue). **b** Confirmation of the expression levels of transfected wild-type and mutants of ELMO1 in COS-7 cells via western blotting using an anti-GFP antibody. GAPDH was used a loading control. **c** Percentages of cells with or without lamellipodia or localised lamellipodia in (**a**) are shown in the graph (n = 3–5 fields with error bars representing SD; ***, p < 0.005 of one-way analysis of variance with Fisher's protected least significant difference test).

~20 °C in a humidified chamber, which was handcrafted using a pipette chip box and moistened paper. First, a 10-µL droplet of Mg-free buffer [20 mM HEPES-NaOH (pH 7.0), 150 mM NaCl] was placed on a PTFE plate, and 1 µL of 0.6 mM lipid mixture [67PC/21PA/6PIP3/6DOGS-NTA Ni (mol %)] dissolved in a 9:1 mixture of chloroform/methanol was gently placed on the buffer droplet and then incubated for 1 h at 25 °C to allow chloroform evaporation. A lipid monolayer formed on the surface of the buffer droplets after evaporation. Next, a carbon-coated cryo-EM grid (Au, R 2/1, 200 mesh; Quantifoil Micro Tools GmbH, Thüringen, Germany) without glow discharge was gently placed on the carbon-coated side down onto the monolayer-formed droplet and incubated again for 1 h at 25 °C. After incubation, the lipid monolayer formed on the droplet surface was transferred to the carbon-coated side of the grid. The lipid membrane-coated grid was allowed to stand on another 10-µL droplet of Mg-free buffer until use. The purified DOCK5•ELMO1 complex was mixed with a two-fold molar excess of Rac1 and RhoG in sample buffer [20 mM HEPES-NaOH (pH 7.0), 150 mM NaCl, and 1 mM MgCl₂], and pre-incubated on ice for at least 1 h.

Afterward, a 10-µL droplet of the pre-incubated protein mixture (final 0.4 µM) was placed on a PTFE plate, and a membrane-coated grid was placed on top of the protein mixture droplet and allowed to stand for 5 min. After the reaction, the grid was washed three times via dipping in a 10-µL droplet of sample buffer, then blotted for 1.5 s in a 100% humidified atmosphere at 4 °C and plunged into liquid ethane using Vitrobot mark IV (Thermo Fisher Scientific). Data acquisition was performed on a Titan Krios G4 electron microscope (Thermo Fisher Scientific) operated at 300 kV and equipped with a K3 direct electron detector (Gatan, Pleasanton, CA, USA) in counting mode. A total of 1931 movies were collected using the EPU software (Thermo Fisher Scientific), including 864 at zero tilt and 1067 acquired at a 30° tilt using stage alpha. Each movie comprised a 4.4-s exposure at a nominal magnification of 64,000×, corresponding to a calibrated pixel size of 1.33 Å/pixel, with a nominal defocus range of −0.8 to −2.0 µm, and was divided into 48 frames. The electron flux rate at the detector was set to 20.1 e⁻/pixel/s, resulting in a total exposure of 50 e⁻/Å² at the specimen.

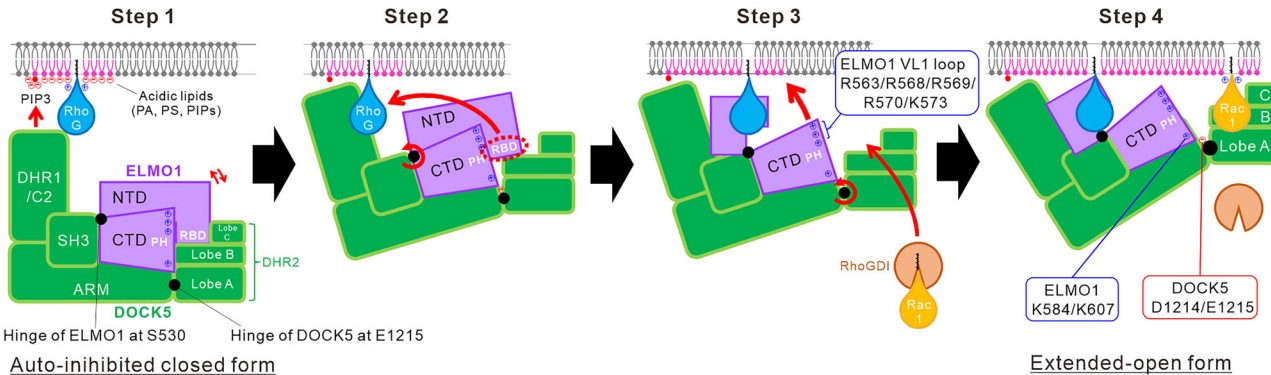

**Fig. 6 | Schematic diagram of conformational changes of the DOCK5•ELMO1 complex on the lipid membrane.** Red arrows indicate the movement of each domain of the DOCK5•ELMO1 complex and Rac1 during the conformational transitions.

## Cryo-EM single-particle analysis

All image processing for the single-particle analysis in this study was performed using cryoSPARC (CryoEM Solutions Inc., Toronto, ON, Canada)[34], a program suite for cryo-EM single-particle analysis. The data processing scheme is shown in Supplementary Fig. 2. The collected movies were motion-corrected using full-frame motion correction with dose weighting, and the parameters of the contrast transfer function were estimated from the motion-corrected micrographs using Patch CTF Estimation. The particles were picked using a Template Picker and extracted using a box size of 360 pixels. Approximately 500,000 extracted particles were classified by repeating Ab-initio Reconstruction and Heterogenous Refinement without the 2D classification process. During this process, classes with higher resolution and more clearly visualized maps corresponding to ELMO1NTD and RhoG were selected. Finally, the map was refined using non-uniform refinement with *C2* symmetry. The resolution of the refined map was 6.98 Å; however, the map corresponding to ELMO1NTD and RhoG was blurred (Supplementary Fig. 3a, b). Therefore, local refinement was performed on the region containing ELMO1NTD and RhoG, resulting in maps of sufficient quality to fit the structural model (Supplementary Fig. 4a, b).

## Model building and refinement

The structural model was constructed using the reported models of the DOCK5•ELMO1 (PDB ID: 8JHK)[15], DOCK5$^{\Delta C}$•ELMO1CTD•Rac1$^{G15A}$ (PDB ID: 7DPA)[16], and ELMO1RBD•RhoG (PDB ID: 7Y4A)[19] complexes. The structural models of DOCK5 and ELMO1 from 8JHK were split into hinge sites (Supplementary Fig. 5a, b). Next, the split models were fitted to the cryo-EM map using UCSF Chimera[35] and manually adjusted using COOT[36]. Rac1$^{G15A}$ from 7DPA and RhoG$^{Q61L}$ from 7Y4A were also fitted to the map using UCSF Chimera. The final model was refined using Phenix[37].

## Lipid–protein interaction assay

A lipid–protein interaction assay was performed using membrane lipid strips (P-6001; Echelon Biosciences) according to Meller's method[38]. All procedures were performed at 25 °C. First, the membrane lipid strips were blocked using blocking buffer [3% w/v bovine serum albumin (BSA) in TBS-T (20 mM Tris-HCl (pH8.0), 150 mM NaCl, 0.1% Tween20)] for 1 h, and then treated with 2 µg/mL of GST-ELMO1CTD in blocking buffer for 1 h. Next, the strips were washed thrice with TBS-T and incubated with primary antibodies [1/2000 diluted anti-GST IgG (SC-138) in blocking buffer] for 1 h. The strips were then washed thrice with TBS-T and incubated with secondary antibodies (1/2000 diluted anti-mouse IgG-HRP in blocking buffer) for 1 h. For detection, the strips were washed thrice with TBS-T after the secondary antibody reaction, immersed in ECL reagent (Chemi-Lumi One Super; Nacalai Tesque), and immediately transferred to a detector (FUSION; Vilber Bio Imaging, Marne-la-Vallée, France) to capture the images.

## Coprecipitation assay using small unilamellar vesicles

The SUVs used in this experiment were prepared by mixing lipids (50 mol% PE and 50 mol% PC, with or without 30 mol% PA) in a 9:1 mixture of chloroform/methanol in a glass tube, followed by drying into a thin film under a nitrogen gas stream. The lipid film was then hydrated with 10% w/w sucrose in 20 mM HEPES-NaOH buffer (pH 7.0) containing 150 mM NaCl, and the resulting suspension was sonicated until it became optically clear. The SUVs (final concentration: 2 mg/mL) were incubated with purified GST-ELMO1CTD (final concentration: 0.2 mg/mL) in 100 µL of 20 mM Tris-HCl (pH 8.0), 150 mM NaCl at 4 °C for approximately 1 h. After the reaction, the mixture was ultracentrifuged at $100,000 \times g$ for 1 h at 4 °C. The pellet was resuspended in the same buffer and ultracentrifuged again at $100,000 \times g$ for 1 h at 4 °C. The final pellet was then resuspended in 100 µL of 20 mM Tris-HCl (pH 8.0), 150 mM NaCl. A volume of 8 µL of this sample was analysed via SDS–PAGE, and the band intensities were quantified using the ImageJ software (National Institutes of Health, Bethesda, MD, USA). Independent experiments were performed three times, and SDS-PAGE was conducted in triplicate for each experiment, resulting in a total of nine data points used for analysis. Statistical analysis was performed using a two-sided *t*-test in Microsoft Excel (Supplementary Data 1).

## In vitro GEF assay using small unilamellar vesicles

The GEF activity assay in the presence of small unilamellar vesicles (SUVs) was performed according to the method described by Robbe et al.[22] with modifications to adapt to the assay conditions for DOCK5. This assay measures fluorescence recovery because of the incorporation of BODIPY FL GTP into Rac1, which dissociates from RhoGDI in the presence of SUV and DOCK5. The purified DOCK5•ELMO1CTD complex (final concentration; 5 nM) and substrate/SUV mixture [final concentration; 2 µM RhoG-DI•Rac1, 2.4 µM BODIPY FL GTP (ThermoFisher Scientific, G12411), 100 µM 78PC/20PA/2PIP3-SUV] were rapidly mixed in assay buffer [20 mM HEPES-NaOH (pH 7.0), 150 mM NaCl, 1 mM MgCl$_2$] and immediately transferred to a 96-well black half-well plate (675077; Greiner, Kremsmünster, Austria). Subsequently, fluorescence was measured at excitation/emission wavelengths of 485 nm/535 nm at 30 °C for 1 h with a 30-s interval using a multi-mode plate reader (SpectraMax iD3; Molecular Devices, San Jose, CA, USA). The observed rate constants ($k_{obs}$), calculated from the measured data through non-linear least-squares fitting of the data using a single exponential rise model in the KaleidaGraph software (Synergy Software, Pleasanton, CA, USA), are shown as GEF activity. Statistical analysis was performed using a two-sided *t*-test in Microsoft Excel (Supplementary Data 2 and Supplementary Data 3).

## Immunofluorescence

The COS-7 cell line (Human Science Research Resources Bank, Tokyo, Japan) was cultured on cell and tissue culture dishes (Nalge Nunc International, Rochester, NY, USA) in DMEM (Nacalai Tesque) containing 10% heat-inactivated foetal bovine serum (FBS) and PenStrep mixture (Thermo

Fisher Scientific) in 5% $CO_2$ at 37 °C. Afterward, the cells were transfected with the respective plasmids using a ScreenFect A kit (FUJIFILM Wako Pure Chemical Corporation, Osaka, Japan) in accordance with the manufacturer's instructions. Briefly, the medium was replaced 4 h post-transfection. Before fixing the cells, the medium containing FBS was replaced with 1 mg/mL BSA (Nacalai Tesque) with or without FBS for 12 h. Under these conditions, attached cells incorporating trypan blue were estimated to be <5% in each experiment. Transfected cells on coverslips were fixed with 4% paraformaldehyde (Nacalai Tesque) or 100% cold methanol (Nacalai Tesque). Next, the cells were blocked with Blocking One (Nacalai Tesque), incubated with primary antibodies [1/500 diluted anti-GFP IgG (MBL, IE4)], pre-loaded with fluorescent dye-conjugated secondary antibodies or fluorescent dye-conjugated chemicals, including phalloidin (Thermo Fisher Scientific), and mounted using Vectashield with a DAPI kit (Vector Laboratories, Burlingame, CA, USA). Fluorescent images were obtained under an FV4000 microscope equipped with a laser-scanning FluoView apparatus (Olympus Corporation, Tokyo, Japan). Multiple images were captured, merged and analysed using the ImageJ software (Supplementary Data 4). To confirm the expression levels of wild-type and mutant ELMO1 in COS-7 cells, western blotting was performed using anti-GFP (MBL, Medical & Biological Laboratories Co., Ltd., Nagoya, Japan; #598, lot#084) and anti-GAPDH (Santa Cruz Biotechnology, Inc., Dallas, TX, USA; #sc-32233, lot#D1125) antibodies.

## Statistics and reproducibility

All experiments were independently performed at least three times. Data are presented as the mean ± standard deviation (SD), as indicated in the figure legends. The statistical methods used are described in the relevant assay sections of the Materials and Methods. Details of the obtained *p*-values are provided in each figure legend.

## Data availability

All structural data were deposited in the Protein Data Bank (PDB) with accession codes EMD-63464, EMD-63477, PDB code 9LX0 and 9LXH. Uncropped and unedited the SDS-PAGE gel and the western blot images are provided in Supplementary Figs. 7−10. All remaining data are available within the Article or the Supplementary Information.

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

## Acknowledgements

The cryo-EM experiments were performed at the RIKEN Yokohama cryo-EM facility. We thank R. Akasaka and T. Uchikubo-Kamo (RIKEN Center for Integrative Medical Sciences) for assistance with data collection and analysis. This work was supported by CREST from Japan Science and Technology Agency (grant number JPMJCR22E3 (M.S.)), and in part by RIKEN Pioneering Projects "Biology of Intracellular Environments" (M.S.).

## Author contributions

Conceptualization: T.S., M.K.N., J.Y., M.S. Methodology: T.S., M.K.N., J.Y., M.S. Investigation: T.S., K.K., Y.I.K., K.H., M.Y., M.K.N., Y.M. Visualization: T.S., J.Y. Funding acquisition: M.S. Project administration: M.S. Supervision: M.S. Writing – original draft: T.S., M.K.N., J.Y. Writing – review & editing: T.S., M.K.N., J.Y., M.S.

## Competing interests

The authors declare no competing interests.
