## [Transparent Peer Review file · Communications Biology]

Conformational alteration of DOCK5•ELMO1 signalosome on lipid membrane

Corresponding Author: Professor Mikako Shirouzu

Version 0:

Reviewer comments:

Reviewer #1

(Remarks to the Author)
Summary of results

This study uncovers a novel regulatory mechanism of DOCK5-mediated activation of Rac1 at the plasma membrane stimulated by ELMO1-RhoG. Using cryo-EM with lipid membrane-coated grids, the authors reveal a previously uncharacterized, membrane-associated conformation in which DOCK5, its activator ELMO1-RhoG, and Rac1 align in a planar arrangement. This new conformation arises from DOCK5 hinge rotation upon interaction with acidic lipids which leads to enhanced ELMO interaction with a membrane. The structural analysis is completed by biochemical and cellular experiments which demonstrate that these membrane-induced conformational changes are critical for the GEF activity of the complex and for triggering downstream signaling pathways.

General assessment

The article is very well written, the introduction is concise and well-structured, providing all necessary background for a clear understanding of the biological context and the rationale behind the study. The results are convincingly presented and well-illustrated. Using single particle analysis cryoEM approach on lipidic monolayer, the authors describe a series of subtle but significant conformational rearrangements within the DOCK5–ELMO1–Rac1–RhoG assembly, leading to a flat membrane-facing configuration. These rearrangements bring new residues, since then unexpected, into proximity with the membrane, including a positively charged tract in ELMO1. The functional importance of these residues is validated through targeted mutagenesis (substitution to uncharged residues) and liposome flotation assays with PS/PA-containing liposomes. The authors go further by testing their hypotheses both in vitro and in cellulo. GEF activity measurements, using Rac1-GDI complexes and fluorescent GTP-bodipy, support the structural findings. Moreover, cell biology experiments in COS-7 cells confirm that the proposed mutations affect Rac activation and downstream signaling, reinforcing the physiological relevance of the proposed structural mechanism. Finally, the manuscript concludes with an elegant regulatory model summarizing the conformational transitions and membrane engagement of the complex.

Minor Points to Address

- activity data: The activity graphs are difficult to interpret due to resolution or formatting issues, making the kobs values nearly illegible. A table of values would help clarify the magnitude of the effects observed. Additionally, the absence of a control for spontaneous nucleotide exchange raises concerns, such a control is necessary to confirm that the observed nucleotide exchange activity is in fact catalyzed by the complex.

-Rac1 membrane recruitment: Does DOCK-ELMO complex exhibit a GEF activity in solution? If the fraction of Rac1 that is membrane-bound in these assays is very low, the measured activity would be averaged with that of cytosolic Rac1, potentially dampening the observable effects of the membrane-interaction defective mutants? Notably, some would expect stronger effect of VL1 mutant on membranes. While no additional experiments are required, this point would benefit from a brief discussion in the manuscript.

-considerations regarding kinetics: Related to previous question, is it known whether membrane recruitment is faster than

GTP exchange? Is membrane binding or conformational reorganization the rate-limiting step, particularly in the presence of ELMO1? If data on these kinetic aspects exist, they should be discussed, or at least acknowledged as an open question.

-discussion: The proposed model is compelling, but the discussion could be expanded to further explore the mechanistic basis of the observed regulation. Is this simply a conformational equilibrium between cytosolic, open, and extended states that enables full DOCK activity on membranes? Or could it represent a bidirectional process, in which after complex initially binds the membrane in a non-productive conformation (incompatible with full activity) through a specific phosphoinositide, then actively reshapes the local membrane environment, potentially inducing lipid sorting or organizing specific microdomains, that in turn promotes conformational rearrangements leading to full activation of the complex? Thus, membrane dynamics would in addition to protein dynamics, contribute to conformational rearrangement. Addressing these possibilities might add depth to the proposed model and could stimulate further investigation in this area.

Conclusion

This is a high-quality and technically impressive study that significantly advances our understanding of Rho GTPase regulation by the DOCK5–ELMO1 complex. The integration of cryo-EM, mutagenesis, liposome-based assays, and cellular validation provides a comprehensive and convincing results.

I particularly appreciate the structural component of the manuscript. The authors successfully adapted a classic, though rarely used in the field, approach, based on lipid monolayer-coated cryoEM grids, to overcome one of the major challenges in structural biology of small GTPases: capturing membrane-bound conformations of dynamic regulatory complexes. This remains a significant technical challenge, particularly for Rho GTPases complexes which prefer flat membrane and are therefore poorly captured with traditional liposome models, especially since, during cryo-EM grid preparation, smaller, highly curved liposomes are more likely to be retained in the thin ice.

Even though the achieved resolution is moderate, the level of structural detail obtained is nonetheless highly impressive. This work not only provides valuable mechanistic insights but also serves as a proof-of-concept, demonstrating the potential of monolayer-based methods to resolve membrane-associated small GTPases complexes using SPA cryoEM approaches. As such, it is likely to inspire further technical improvements, hopefully contributing, alongside other emerging strategies, to overcoming the current bottlenecks in resolving membrane-associated protein complexes.

To conclude: the manuscript is well-suited for publication in a high-impact journal. Minor clarifications, particularly regarding GEF activity controls and mechanistic discussion, would further enhance its clarity and impact.

Agata Nawrotek

Reviewer #2

(Remarks to the Author)

In this manuscript, Shinoda and colleagues present an elegant structural, biochemical and cell biology investigation of the ELMO-DOCK5 complex in the context of membranes, in the presence of RhoG and Rac1. To achieve this, the authors developed an innovative cryo-EM approach using lipid-coated grids, enabling them to capture the ELMO-DOCK5 complex in a membrane-mimicking environment. This technical advancement allows visualization of new features of the complex, including a flattened, membrane-aligned conformation.

A key finding is the identification of a basic residue-rich variable loop (VL1) within the ELMO-PH domain that mediates membrane association. Mutating these residues leads to a modest (~10%) reduction in GEF activity in Rac1-coated liposome assays, supporting their functional relevance. Additionally, the authors map hinge regions in ELMO and DOCK5 that enable the complex to transition from the open conformation to the more elongated/flatten one. These subtle but important structural changes appear critical for aligning the GEF complex on the membrane and optimizing its activity.

Collectively, this manuscript significantly advances our mechanistic understanding of DOCK-A and DOCK-B GEFs function within the membrane microenvironment. It builds on previous structural work from this group but extends it in meaningful ways. I believe the study is timely and impactful, and I recommend publication pending minor revisions.

Minor Comments

1. **Structural Overview:** While the study is clearly written for readers familiar with DOCK family GEFs, it would benefit from a brief but clear visual summary of ELMO-DOCK structural domains early in the manuscript. For instance, adding a domain map or cartoon model to Figure 2 could improve accessibility for a broader audience and help contextualize the structural transitions described later.
2. **VL1–Lipid Interaction:** The PIP-strip assay provides initial evidence that the VL1 region of ELMO binds to lipids. However, a complementary biochemical assay (e.g., liposome co-sedimentation using the same liposomes as in the GEF assay) would strengthen this conclusion. Alternatively, isothermal titration calorimetry (ITC) could be considered, though the authors may wish to discuss whether the interaction is too weak for this method. If additional experiments are not feasible, a brief discussion of these limitations would be useful.
3. **Cellular Assays – Additional Controls:** The cell biology data are generally convincing, but a few controls would solidify the conclusions. Specifically:

- (i) Please include a condition with empty vector control to evaluate baseline actin organization.
- (ii) Confirmation of equal expression levels of wild-type and mutant proteins via Western blot would reinforce the interpretation that observed differences are due to functional, not expression-level, effects.

4. Model of Activation Mechanism: The model in Figure 5 proposing PIP3–DHR-1 interaction as an initiating step for complex opening is intriguing. However, it remains unclear whether this interaction alone is sufficient to drive the transition from closed to open conformation. The authors should clarify whether they propose an equilibrium model (where PIP3 binding shifts the equilibrium) or whether additional interactions (e.g., with RhoG or Bai1) are required to stabilize the open conformation. A more explicit mechanistic model—perhaps in the form of an updated schematic—would greatly enhance the impact of the discussion.

5. Language and Style: The manuscript would benefit from careful proofreading to address minor grammatical issues and improve overall clarity. For instance, phrases such as “thrice” should be replaced with “three times,” and similar stylistic adjustments should be made throughout the methods and results sections.

In summary, this is a carefully executed and insightful study that makes a valuable contribution to the field. With minor revisions to improve clarity and reinforce a few experimental claims, it will be well-suited for publication.

Version 1:

Reviewer comments:

Reviewer #1

(Remarks to the Author)

Dear Authors,

Many thanks for your thorough responses and the improvements to the manuscript. The revisions are clear, address the points raised, and substantially strengthen the paper. I find the work novel and highly important for the field, and I strongly recommend the manuscript for publication.

Reviewer #2

(Remarks to the Author)

I wish to congratulate the authors for their investment in answering reviewers' comments. The authors have now addressed all of my comments adequately.

Rebuttal Letter

Reviewer #1 (Remarks to the Author):

Summary of results

This study uncovers a novel regulatory mechanism of DOCK5-mediated activation of Rac1 at the plasma membrane stimulated by ELMO1-RhoG. Using cryo-EM with lipid membrane-coated grids, the authors reveal a previously uncharacterized, membrane-associated conformation in which DOCK5, its activator ELMO1-RhoG, and Rac1 align in a planar arrangement. This new conformation arises from DOCK5 hinge rotation upon interaction with acidic lipids which leads to enhanced ELMO interaction with a membrane. The structural analysis is completed by biochemical and cellular experiments which demonstrate that these membrane-induced conformational changes are critical for the GEF activity of the complex and for triggering downstream signaling pathways.

General assessment

The article is very well written, the introduction is concise and well-structured, providing all necessary background for a clear understanding of the biological context and the rationale behind the study.

The results are convincingly presented and well-illustrated. Using single particle analysis cryoEM approach on lipidic monolayer, the authors describe a series of subtle but significant conformational rearrangements within the DOCK5–ELMO1–Rac1–RhoG assembly, leading to a flat membrane-facing configuration. These rearrangements bring new residues, since then unexpected, into proximity with the membrane, including a positively charged tract in ELMO1. The functional importance of these residues is validated through targeted mutagenesis (substitution to uncharged residues) and liposome flotation assays with PS/PA-containing liposomes. The authors go further by testing their hypotheses both in vitro and in cellulo. GEF activity measurements, using Rac1-GDI complexes and fluorescent GTP-bodipy, support the structural findings. Moreover, cell biology experiments in COS-7 cells confirm that the proposed mutations affect Rac activation and downstream signaling, reinforcing the physiological relevance of the proposed structural mechanism.

Finally, the manuscript concludes with an elegant regulatory model summarizing the conformational transitions and membrane engagement of the complex.

Minor Points to Address

Comment 1: *Activity data: The activity graphs are difficult to interpret due to resolution or formatting issues, making the kobs values nearly illegible. A table of values would help clarify the magnitude of the effects observed. Additionally, the absence of a control for spontaneous*

nucleotide exchange raises concerns, such a control is necessary to confirm that the observed nucleotide exchange activity is in fact catalyzed by the complex.

Response: We thank the reviewer for the suggestion. In response, we performed a control experiment to assess spontaneous nucleotide exchange. We provide the obtained data as Fig. 4d and describe the results in lines 29–36 on p.5. In addition, we have included the k_{obs} values in the GEF assay data in Fig. 4d and Fig.4e. In our assay using the RhoGDI•Rac1 complex as the substrate, we detected no nucleotide exchange when either the DOCK5•ELMO1 GEF complex or the SUV lipid membrane was omitted.

Revised Fig. 4d and Fig. 4e. **d**, GEF assay using RhoGDI•Rac1 to evaluate the GEF activity of DOCK5•ELMO1CTD in the absence and presence of SUVs. Graph shows the average of three experiments, with error bars representing standard deviation (SD). **e**, Assessment of GEF activity of DOCK5•ELMO1CTD mutants. GEF activity was assessed by measuring the exchange of GDP for Bodipy-GTP on Rac1 in the presence of SUVs composed of 78PC/20PA/2PIP3 (mol%). Data are presented as the mean \pm SD ($n = 3$; unpaired two-sided Student's t -test, *** $p < 0.005$).

Comment 2: *Rac1 membrane recruitment: Does DOCK-ELMO complex exhibit a GEF activity in solution? If the fraction of Rac1 that is membrane-bound in these assays is very low, the measured activity would be averaged with that of cytosolic Rac1, potentially dampening the observable effects of the membrane-interaction defective mutants? Notably, some would expect stronger effect of VL1 mutant on membranes. While no additional experiments are required, this point would benefit from a brief discussion in the manuscript.*

Response: We are grateful for this insightful point. In our GEF assays, we used the RhoGDI•Rac1 complex as the substrate. RhoGDI binds tightly to GDP-bound Rho GTPases, including Rac1, and suppresses GDP exchange; previous studies have indicated that both GEF and a lipid membrane are essential for the dissociation of RhoGDI from GDP-bound Rho GTPases. No nucleotide exchange was observed when either the DOCK5•ELMO1 GEF complex or the SUV lipid membrane was omitted as shown in the newly added Fig. 4d. Thus, we did not observe spontaneous dissociation of RhoGDI from Rac1 or nucleotide exchange in solution. Therefore, Fig. 4d and Fig. 4e present only the

GEF activity of DOCK5•ELMO1 localized on membranes. Accordingly, for the membrane-defective VL1 mutant interaction analyzed in Fig. 4e using the same assay, we considered that the observed effects also reflected events occurring on membranes. We have added this explanation in lines 37–39 on p.5.

Comment 3: *Considerations regarding kinetics: Related to previous question, is it known whether membrane recruitment is faster than GTP exchange? Is membrane binding or conformational reorganization the rate-limiting step, particularly in the presence of ELMO1? If data on these kinetic aspects exist, they should be discussed, or at least acknowledged as an open question.*

Response: Thank you for highlighting these kinetic aspects. We did not perform time-resolved measurements that would allow us to compare the rates of membrane recruitment and nucleotide exchange. In our assay using the RhoGDI•Rac1 substrate, nucleotide exchange was detected only when both the DOCK5•ELMO1 complex and SUVs were present, indicating that membrane engagement is required. However, under the experimental conditions used, we could not distinguish whether membrane binding or subsequent conformational reorganization (e.g., adoption of the extended-open form) was rate-limiting, particularly in the presence of ELMO1. We have added an interpretation of Fig. 4e in the Results section (lines 5–9 on p.6) and included a corresponding discussion in the Discussion section (lines 7–18 on p.8). Furthermore, we have added a statement in Conclusion (lines 6–10 on p.9) to acknowledge this limitation and to suggest future experiments that could resolve the sequence and rate-limiting step.

Comment 4: *Discussion: The proposed model is compelling, but the discussion could be expanded to further explore the mechanistic basis of the observed regulation. Is this simply a conformational equilibrium between cytosolic, open, and extended states that enables full DOCK activity on membranes? Or could it represent a bidirectional process, in which after complex initially binds the membrane in a non-productive conformation (incompatible with full activity) through a specific phosphoinositide, then actively reshapes the local membrane environment, potentially inducing lipid sorting or organizing specific microdomains, that in turn promotes conformational rearrangements leading to full activation of the complex? Thus, membrane dynamics would in addition to protein dynamics, contribute to conformational rearrangement. Addressing these possibilities might add depth to the proposed model and could stimulate further investigation in this area.*

Response: We thank the reviewer for this insightful comment. In response, we have revised the schematic diagram in Fig. 6 to improve clarity. Corresponding additions and modifications have also been made in the Results section (lines 35–40 on p.6 and lines 4–14 on p.7) as well as in the Discussion section (lines 19–20, lines 22–24, and lines 29–39 on p.8). Furthermore, we have cited and added three supporting references, as shown below:

“Remorino, A. et al., *Cell Reports*, 2017, <https://doi.org/10.1016/j.celrep.2017.10.069>”

“Chuang, T. H. et al., *J. Biol. Chem.*, 1993, [https://doi.org/10.1016/S0021-9258\(19\)74301-4](https://doi.org/10.1016/S0021-9258(19)74301-4)”

“Garcia-Mata, R. et al., *Nat. Rev. Mol. Cell Biol.*, 2011, <https://doi.org/10.1038/nrm3153>”

Through the structural analysis of the extended-open form and supporting validation experiments, we have demonstrated that in the signalling pathway involving the DOCK5•ELMO1 complex, activation of Rac1 by GEF activity alone is insufficient. Instead, the interaction between the PH domain of ELMO1 in the extended-open form and the lipid membrane is critical. Although we could not establish this directly in the present study, we speculate that the interaction between the PH domain and the membrane promotes the formation of a local membrane domain enriched in acidic lipids, which in turn may facilitate the clustering of activated Rac1, which has been reported to be involved in downstream signalling.

Revised Fig. 6 | Schematic diagram of conformational changes of the DOCK5•ELMO1 complex on the lipid membrane. Red arrows indicate the movement of each domain of the DOCK5•ELMO1 complex and Rac1 during the conformational transitions.

Conclusion

This is a high-quality and technically impressive study that significantly advances our understanding of Rho GTPase regulation by the DOCK5-ELMO1 complex. The integration of cryo-EM, mutagenesis, liposome-based assays, and cellular validation provides a comprehensive and convincing results.

I particularly appreciate the structural component of the manuscript. The authors successfully adapted a classic, though rarely used in the field, approach, based on lipid monolayer-coated cryoEM grids, to overcome one of the major challenges in structural biology of small GTPases: capturing membrane-bound conformations of dynamic regulatory complexes. This remains a significant technical challenge, particularly for Rho GTPases complexes which prefer flat membrane and are therefore poorly captured with traditional liposome models, especially since, during cryo-EM grid preparation, smaller, highly curved liposomes are more likely to be retained in the thin ice.

Even though the achieved resolution is moderate, the level of structural detail obtained is nonetheless highly impressive. This work not only provides valuable mechanistic insights but also serves as a proof-of-concept, demonstrating the potential of monolayer-based methods to resolve membrane-associated small GTPases complexes using SPA cryoEM approaches. As such, it is likely to inspire further technical improvements, hopefully contributing, alongside other emerging strategies, to overcoming the current bottlenecks in resolving membrane-associated protein complexes.

To conclude: the manuscript is well-suited for publication in a high-impact journal. Minor clarifications, particularly regarding GEF activity controls and mechanistic discussion, would further enhance its clarity and impact.

Reviewer #2 (Remarks to the Author):

In this manuscript, Shinoda and colleagues present an elegant structural, biochemical and cell biology investigation of the ELMO-DOCK5 complex in the context of membranes, in the presence of RhoG and Rac1. To achieve this, the authors developed an innovative cryo-EM approach using lipid-coated grids, enabling them to capture the ELMO-DOCK5 complex in a membrane-mimicking environment. This technical advancement allows visualization of new features of the complex, including a flattened, membrane-aligned conformation.

A key finding is the identification of a basic residue-rich variable loop (VL1) within the ELMO-PH domain that mediates membrane association. Mutating these residues leads to a modest (~10%) reduction in GEF activity in Rac1-coated liposome assays, supporting their functional relevance. Additionally, the authors map hinge regions in ELMO and DOCK5 that enable the complex to transition from the open conformation to the more elongated/flattened one. These subtle but important structural changes appear critical for aligning the GEF complex on the membrane and optimizing its activity.

Collectively, this manuscript significantly advances our mechanistic understanding of DOCK-A and DOCK-B GEFs function within the membrane microenvironment. It builds on previous structural work from this group but extends it in meaningful ways. I believe the study is timely and impactful, and I recommend publication pending minor revisions.

Minor Comments

Comment 1: Structural Overview: *While the study is clearly written for readers familiar with DOCK family GEFs, it would benefit from a brief but clear visual summary of ELMO-DOCK structural domains early in the manuscript. For instance, adding a domain map or cartoon model*

to Figure 2 could improve accessibility for a broader audience and help contextualize the structural transitions described later.

Response: We thank the reviewer for the suggestion. In response, we included domain maps in Fig. 2a. Accordingly, we made minor adjustments to the font sizes and layout of panels b–d to improve visual clarity.

Revised Fig. 2 | Cryo-EM structure of DOCK5•ELMO1 complex with Rac1 and RhoG on a lipid membrane. **a**, Domain organisation of human DOCK5, ELMO1, and two Rho-GTPases, Rac1 and RhoG, and of the protein constructs used in each experiment in this study (black bar). DHR, Dock homology region; ARM, armadillo repeat domain; RBD, Ras-binding domain; EID, ELMO inhibitory domain; ELM, ELMO domain. **b–d**, Cryo-EM single-particle analysis of the DOCK5•ELMO1 complex bound to Rac1 and RhoG. Shown are the 2D class average (**b**), cryo-EM density maps displayed at a contour level of 0.12 (**c**), and the cryo-EM structural model of the extended-open form of the DOCK5•ELMO1 complex associated with Rac1 and RhoG, generated from the maps in (**c**) and compared with the auto-inhibited closed form of the DOCK5•ELMO1 complex (PDB: 8JHK) (**d**). In panels (**c**) and (**d**), the upper and lower images represent front and top views, respectively.

Comment 2: VL1–Lipid Interaction: The PIP-strip assay provides initial evidence that the VL1 region of ELMO binds to lipids. However, a complementary biochemical assay (e.g., liposome co-sedimentation using the same liposomes as in the GEF assay) would strengthen this conclusion. Alternatively, isothermal titration calorimetry (ITC) could be considered, though the authors may wish to discuss whether the interaction is too weak for this method. If additional experiments are not feasible, a brief discussion of these limitations would be useful.

Response: We thank the reviewer for the helpful suggestion. We performed the suggested liposome coprecipitation assay and included the results in Fig. 4b and Fig. 4c, with the corresponding text provided in lines 8–18 on p.5 and in the “Coprecipitation assay” section of the Materials and Methods (lines 4–21 on p.20). Apart from including the coprecipitation assay data, the immunofluorescence data originally provided in Fig. 4b and Fig. 4c are now presented in Fig. 5. Consequently, the original Fig. 5 (schematic diagram of conformational changes) has been renumbered as Fig. 6. Furthermore, the PIP-strip assay data (originally Fig. 3d) were combined with the coprecipitation assay data into the new Fig. 4. Following the removal of the PIP-strip assay data, the title of Fig. 3 was revised to better reflect its current content, and the figure layout and font sizes were adjusted to improve visual clarity. In line with these rearrangements, the title of Fig. 4 has been revised accordingly. By comparing wild-type and VL1-mutant ELMO1CTD, we assessed the effect of phosphatidic acid (PA) on the amount of coprecipitated protein. The wild type showed increased coprecipitation in the presence of PA, whereas this effect was attenuated in the VL1 mutant. These data support that the PH domain of ELMO1CTD interacts with lipid membranes via basic residues within VL1.

Revised Fig. 3 | Interactions due to the conformational change of the complex formed by DOCK5•ELMO1 and two Rho-GTPases in the extended-open form. a, Structural models of the closed and extended-open forms of the complex superimposed on the DHR-2 domain. Movements of the α -helix (residues 1186–1211) at the C-terminus of the ARM domain, which connects to the DHR-2 domain in DOCK5 and serves as the epicentre of the conformational change, are indicated by red dashed lines. **b,** Conformational change in the region highlighted by the orange box in (a) and its effect on the interaction between DOCK5 and ELMO. The transition to the extended-open form brings the acidic residues D1214 and E1215 in the hinge region of DOCK5 (between the ARM and DHR-2 domains) into close proximity with the basic residues K584 and K607 in the PH domain of ELMO1. **c,** PH domain of ELMO1 oriented toward the lipid membrane in the extended-open form. Basic residues on the VL1 loop of the PH domain are positioned close to the lipid membrane.

Revised Fig. 4a–c. a, Lipid–protein interaction assay for ELMO1CTD. Proteins were detected using an anti-GST antibody. **b**, Coprecipitation assay using small unilamellar vesicles (SUVs). ELMO1CTD that coprecipitated with SUVs via centrifugation at $100,000 \times g$ were detected using SDS–PAGE followed by Coomassie Brilliant Blue (CBB) staining. **c**, Densitometric analysis of the co-precipitated ELMO1CTD shown in **b**. Data are presented as the mean \pm SD ($n = 9$; unpaired two-sided Student’s *t*-test, $***p < 0.005$, $**p < 0.01$, $*p < 0.05$).

Comment 3: Cellular Assays – Additional Controls: The cell biology data are generally convincing, but a few controls would solidify the conclusions. Specifically:

- (i) Please include a condition with empty vector control to evaluate baseline actin organization.
- (ii) Confirmation of equal expression levels of wild-type and mutant proteins via Western blot would reinforce the interpretation that observed differences are due to functional, not expression-level, effects.

Response: We are thankful for the suggestion. In response, we added immunofluorescence images of COS-7 cells transfected with an empty vector control in Fig. 5a. We also confirmed the expression levels of wild-type and mutant ELMO1 via western blotting and included these results in Fig. 5b, with corresponding descriptions in lines 10–27 on p.6 and in the “Immunofluorescence” section of the Materials and Methods (lines 16–19 on p.21).

Revised Fig. 5 | Mutated ELMO1 decreases localised filamentous actin formation. **a**, COS-7 cells were transfected with a plasmid encoding DOCK5 with or without wild-type or mutant ELMO1 and stimulated with 10% FBS (FBS+) or serum-free medium (FBS-). Transfected cells were stained using an anti-GFP antibody (to detect ELMO1) pre-loaded with green fluorescence dye-conjugated secondary antibody and red fluorescence dye-conjugated phalloidin (to detect filamentous actin). Cells were finally mounted using a DAPI-containing mounting medium (blue). **b**, Confirmation of the expression levels of transfected wild-type and mutants of ELMO1 in COS-7 cells via western blotting using an anti-GFP antibody. GAPDH was used a loading control. **c**, Percentages of cells with or without lamellipodia or localised lamellipodia in (a) are shown in the graph ($n = 3-5$ fields; ***, $p < 0.005$ of one-way analysis of variance with Fisher's protected least significant difference test).

Comment 4: Model of Activation Mechanism: The model in Figure 5 proposing PIP3–DHR-1 interaction as an initiating step for complex opening is intriguing. However, it remains unclear whether this interaction alone is sufficient to drive the transition from closed to open conformation. The authors should clarify whether they propose an equilibrium model (where PIP3 binding shifts the equilibrium) or whether additional interactions (e.g., with RhoG or Bai1) are required to stabilize the open conformation. A more explicit mechanistic model—perhaps in the form of an updated schematic—would greatly enhance the impact of the discussion.

Response: We thank the reviewer for this insightful comment and suggestion. In response, we revised the text of the “schematic diagram” section in lines 40–46 on p.6 to clarify the recruitment of the DOCK5•ELMO1 complex to membranes and its conformational changes on the membrane. We propose that PIP3 recruits the DOCK5•ELMO1 complex via its interaction with the DOCK5 DHR-1 domain but does not directly drive the conformational transition to the extended-open form. We suggest that RhoG contributes to this transition by retaining the ELMO1 N-terminal domain (NTD) near the membrane after its dissociation from DHR-2. BAI1 likely interacts with the ELMO1 NTD at a site distinct from that of RhoG and, similar to RhoG, may promote the transition by retaining the dissociated NTD near the membrane.

Comment 5: *Language and Style: The manuscript would benefit from careful proofreading to address minor grammatical issues and improve overall clarity. For instance, phrases such as “thrice” should be replaced with “three times,” and similar stylistic adjustments should be made throughout the methods and results sections.*

Response: We thank the reviewer for the insightful comment. We carefully proofread the entire manuscript, corrected minor grammatical issues, and improved its overall clarity. Additionally, the sentences in lines 25–31 on p.7, were moved from the “schematic diagram” section and reworded to improve readability. The sentences in lines 2–7 on p.5, and lines 39–43 on p.8, have been reworded to enhanced clarity and accuracy. To improve precision, we have also revised the description of data acquisition in the Materials and Methods (line 2, lines 34–36 and lines 41–46 on p.18; lines 1–4 on p.19; and lines 39–40 and line 43 on p.20). Minor typographical errors throughout the Materials and Methods were collected as well.

In summary, this is a carefully executed and insightful study that makes a valuable contribution to the field. With minor revisions to improve clarity and reinforce a few experimental claims, it will be well-suited for publication.